# Revised Minoan eruption volume as benchmark for large volcanic eruptions

Jens Karstens [1] ✉, Jonas Preine [2], Gareth J. Crutchley [1], Steffen Kutterolf[1], Willem G. M. van der Bilt [3], Emilie E. E. Hooft [4], Timothy H. Druitt[5], Florian Schmid[1,8], Jan Magne Cederstrøm [3], Christian Hübscher [2], Paraskevi Nomikou[6], Steven Carey[7], Michel Kühn [1], Judith Elger [1] & Christian Berndt [1]

Despite their global societal importance, the volumes of large-scale volcanic eruptions remain poorly constrained. Here, we integrate seismic reflection and P-wave tomography datasets with computed tomography-derived sedimentological analyses to estimate the volume of the iconic Minoan eruption. Our results reveal a total dense-rock equivalent eruption volume of $34.5 \pm 6.8$ km³, which encompasses $21.4 \pm 3.6$ km³ of tephra fall deposits, $6.9 \pm 2$ km³ of ignimbrites, and $6.1 \pm 1.2$ km³ of intra-caldera deposits. $2.8 \pm 1.5$ km³ of the total material consists of lithics. These volume estimates are in agreement with an independent caldera collapse reconstruction ($33.1 \pm 1.2$ km³). Our results show that the Plinian phase contributed most to the distal tephra fall, and that the pyroclastic flow volume is significantly smaller than previously assumed. This benchmark reconstruction demonstrates that complementary geophysical and sedimentological datasets are required for reliable eruption volume estimates, which are necessary for regional and global volcanic hazard assessments.

Explosions, tephra fallout, pyroclastic flows, lahars, and tsunamis generated by volcanic eruptions have a profound impact on society and the environment, while large-scale events like the 1815 eruption of Tambora can even alter Earth's climate[1,2]. Despite the global societal impact of eruptions of similar magnitude, little is known about fundamental eruption parameters such as volume or mass-partitioning between pyroclastic flows and tephra fall[2], thereby impeding hazard assessment. The ~1600 BCE Minoan eruption of Santorini was one of the largest in the Holocene[3] and, owing to its cultural relevance, has been the focus of volcanological and archaeological research since the 19th century. Eruption volume estimates are based either on constraining the caldera collapse volume or on mapping the eruptive products. Both approaches have limitations, challenges, and associated errors. Caldera collapse volume estimates require a comparison between pre- and post-collapse topographies, which necessitates a detailed understanding of the intra-caldera stratigraphy. However, calderas are often multicyclic and subsequent filling and volcanism can change the primary dimensions, making it difficult to assign precise volumetric changes to a specific eruption[2]. Eruption volume estimates are based on mapping of eruptive products inside and outside of the caldera and require a combination of techniques dedicated to specific parts and scales of deposits. While cm-thick tephra layers cannot be resolved with geophysical methods, sediment coring only reaches the surface of proximal ignimbrites and caldera infill deposits. Determination of the volume of large-scale volcanic events is also complicated by the fact that eruptive products are

[1]GEOMAR Helmholtz Centre for Ocean Research Kiel, Kiel, Germany. [2]University of Hamburg, Institute of Geophysics, Hamburg, Germany. [3]Department of Earth Science and Bjerknes Centre for Climate Research, University of Bergen, Bergen, Norway. [4]Department of Earth Science, University of Oregon, Eugene, OR, USA. [5]Laboratoire Magmas et Volcans, Université Clermont Auvergne, OPGC, CNRS, IRD, F-63000 Clermont-Ferrand, France. [6]National and Kapodistrian University of Athens, Department of Geology and Geoenvironment, Athens, Greece. [7]University of Rhode Island, Kingston, RI, USA. [8]Present address: K.U.M Umwelt und Meerestechnik Kiel GmbH, Germany. ✉e-mail: jkarstens@geomar.de

transported over vast areas and subsequently affected by remobilisation or masking by later eruptive activity[2]. Together, these challenges result in considerable uncertainty regarding the volume of the Minoan eruption, with estimates ranging between 19 and 86 km³ dense-rock equivalent (DRE)[3,4].

Despite the volume uncertainty, the temporal evolution of the Minoan eruption is one of the best understood of all major Holocene eruptions. It started with precursory explosions some days to weeks before the onset of the main eruption sequence[4–6], which was subdivided into four phases beginning with subaerial Plinian discharge (Phase 1). After several hours, the eruption vent migrated into a pre-existing flooded caldera, where the interaction with sea-water caused violent phreatomagmatic explosions[4–6] and the emplacement of pyroclastic surges (Phase 2), as well as low-temperature pyroclastic flows (Phase 3). This phreatomagmatic activity probably formed a tuff cone that filled up the pre-existing caldera, allowing the low-temperature pyroclastic flows to spill over onto Santorini's flanks[3,7]. Finally, as the tuff cone grew, the connection to the sea was sealed off and water evaporated. Large volumes of hot, fluidised pyroclastic flows ran over the island's slopes forming depositional fans (Phase 4), while caldera collapse during or after the eruption formed the Santorini archipelago's present-day topography[3,6,8]. The entrance of pyroclastic flows during Phases 3 and 4 into the sea may have triggered tsunamis that impacted coasts around the Aegean Sea[9,10]. Accumulations of deposits from these pyroclastic flows (termed ignimbrites) occur in the submarine basins surrounding Santorini.

In this study, we reassess the volume of the Minoan eruption with unmatched accuracy by combining high-resolution seismic reflection profiles, seismic P-wave (V_p) tomography, and X-ray computed tomography (CT) scans of a selection of marine sediment cores to better

constrain the volume of distal tephra, proximal ignimbrite, and intra-caldera deposits. We reconstruct the volume of the caldera collapse based on previously published constraints and compare this estimate with the eruption products, thus providing one of the best constrained volume assessment for a major (>6 magnitude) explosive volcanic eruption.

## Results

### Interpretation of Minoan fall deposits in marine sediment cores
We identified Minoan eruption fallout tephra in 41 gravity cores recovered during research cruise POS513 in 2017 based on their unique geochemical signature (see Kutterolf et al., 2021a)[11–13]. Guided by their visual appearance in photographic scans, sedimentological properties, and fabric in X-ray CT-scans, we define three subunits within the Minoan deposits. This tripartition is clearly demonstrated (and representative of the other cores) in the 50 cm-thick sequence of alternating Minoan ash and lapilli layers of core POS513-20, located ~27 km from the eruptive centre (Fig. 1a). The lower subunit consists of 1–6 cm thick beds comprising subangular to angular pumice clasts and a low lithic content. Grain size variations are clearly illustrated by grayscale variations in the CT-scan and beds are generally continuous (Fig. 1d). A distinct colour change in the photographic scan and a change of the CT-grayscale values mark the boundary with the middle subunit (Fig. 1c), which consists mainly of lithic-rich, fine-grained, and laminated deposits with few subrounded to rounded pumice clasts. The CT-scan of the middle unit shows indications of cross-bedding within the partly discontinuous and inclined strata (Fig. 1b). The upper unit consists of fine-grained, partly disturbed material, which lacks crystals and lithics. In the CT-scan, the upper unit shows no internal structure, aside from bioturbation, and has an irregular boundary at

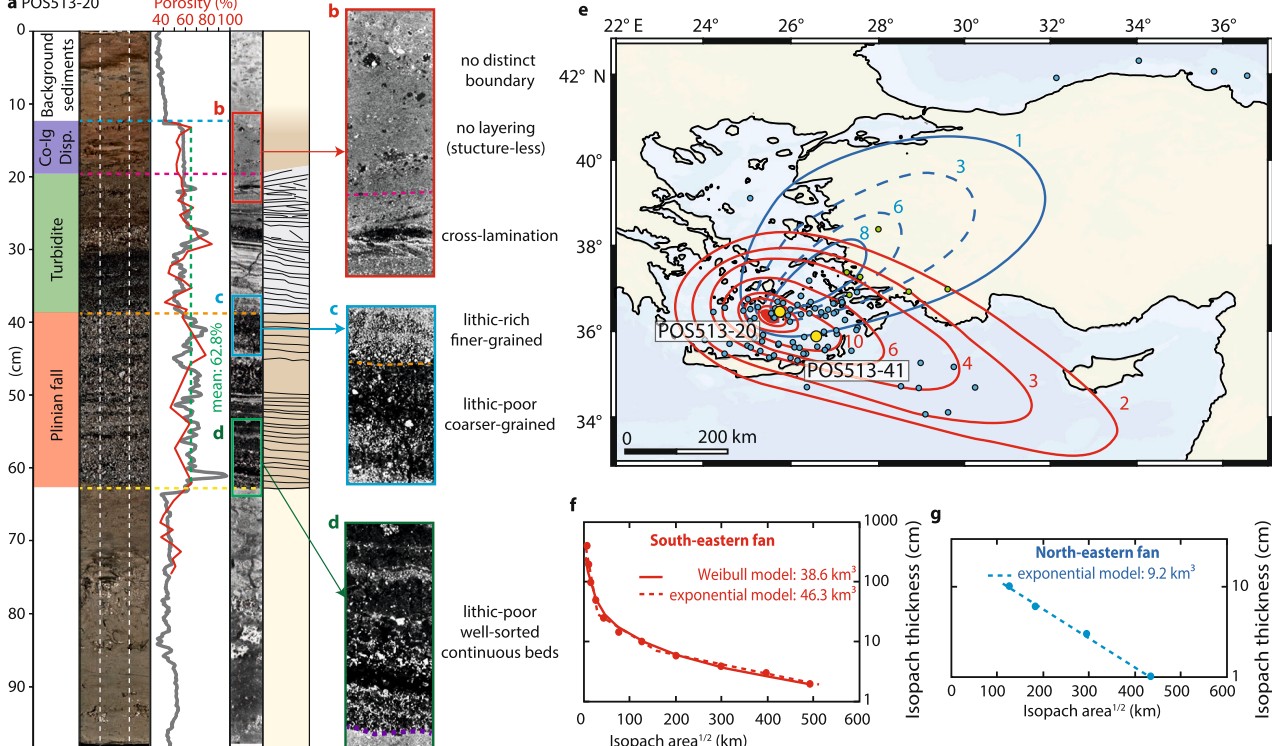

**Fig. 1 | Sedimentological analysis of marine gravity core POS513-20.**
**a** Photographic scans, porosity measurements, X-ray CT-scan, and stratigraphic interpretation. Co-Ig Disp. refers to the dispersed co-ignimbrite layer. **b**–**d** Enlargements of key intervals of the X-ray CT-scan showing differences between the deposits' subunits. **e** Isopach maps (cm) for tephra layer 1 (red; Plinian phase) and tephra layer 2 (blue; co-ignimbrite phase). Bathymetric data: GMRT-Global Multi-Resolution Topography. Dots are sediment core locations; yellow dots are cores POS513-20 and POS513-41. **f** Deposit thinning curve for the lower tephra layer (Plinian phase). **g** Deposit thinning curve for upper tephra layer (co-ignimbrite phase).

the base and a diffuse one at the top, which correlates to a distinct porosity decrease (Fig. 1b).

Density measurements as well as the grayscale values in the CT-scan indicate pronounced porosity variations within the lower and middle subunits. To investigate this variability, we integrated physical grain size and density measurements with CT data as detailed in Supplementary Note 2 to obtain CT-derived porosity estimates, which result in 100 μm resolution down-core porosity information. As shown in Fig. 1a, both CT-derived and physically measured density estimates capture the same trends in the scanned segments, which we attribute to down-core porosity changes suggested by the strong anti-correlation between density and measured particle size (Supplementary Note 2). This means that down-core CT grayscale data can be used to refine down-core density and porosity calculations at 500 μm intervals, and can then be transformed into porosity estimates using solid density values for Minoan tephra[12,13] (Supplementary Data 1). Based on this approach, the lower subunit has a mean porosity of 64.3 ± 8.4%, the middle subunit has a mean porosity of 59.5 ± 6.4%, the upper unit shows a rather constant porosity of 58.2 ± 0.9%, and the entire Minoan sequence has a mean porosity of 62.9 ± 7.6% (Fig. 1a). The observed tripartition is present in several other cores east of Santorini, including the more distal core POS513-41 (Supplementary Fig. S3), where the analysis resulted in a CT-derived mean porosity of 54.9 ± 8%.

Due to its continuous nature of normally graded beds, the angularity of clasts, and the low lithic content, we interpret the lower subunit as Plinian fallout from Phase 1 that was deposited by particles settling through the water column. The cross- and inclined-bedding, the high lithic content, and the non-continuous layering of the middle subunit indicate a more energetic emplacement. Therefore, we interpret the middle subunit as the result of pyroclastic flows (Phases 2–4) that entered the sea and continued as seafloor-bound turbidity currents. Due to its fine grain size, lack of lithics and minerals, and structureless fabric in the CT-scans, we interpret the upper subunit as co-ignimbrite tephra fall deposits or dispersed tephra, mostly from eruptive Phase 4. However, the middle unit can only be identified in cores obtained from a channel system extending east from Santorini (Supplementary Note 1), which agrees with its interpretation as turbidites. In cores lacking the middle unit, the boundary between the lower and upper units is characterised by a distinct change in colour and locally by grain size and angularity (Supplementary Note 1).

### Quantification of marine tephra fall volume

Guided by the described criteria, we identified and measured the thickness of the Plinian layer, the turbidite layer and the co-ignimbrite/dispersed layer, which together form the tephra deposits of the Minoan eruption. Combined with more distal thickness measurements from the region (Supplementary Data 2), we compiled isopach maps for the Plinian (red in Fig. 1e) and the co-ignimbrite/dispersed tephra layers (blue in Fig. 1e; Supplementary Fig. S2). The thickness distribution of the Plinian layer (lower subunit) agrees well with the thinning trend seen in more distal sediment cores from the Eastern Mediterranean[11] and results in isopachs with a southeast-oriented deposit lobe. The co-ignimbrite/dispersed layer (upper subunit) follows the trend of thickness constraints from Aegean Islands, mainland Turkey and the Black Sea, defining a northeast-oriented lobe (blue in Fig. 1e). This confirms that the tephra layers were affected by different prevailing wind directions, which has already been documented for the evolution of the first phase of the Minoan eruption[14]. Compared to the southeast-oriented lobe, there are fewer data points available for the northeast-oriented deposit lobe, with a data gap in north-western Turkey. The 3 mm and 6 mm isopachs were interpolated based on the few available distal occurrences in the Black Sea and in the northern Aegean Sea (indicated by dashed lines, implying less certainty, in Fig. 1e). There was no information about potential partitioning within

Minoan tephra deposits for analyses from earlier studies. However, most available data points could be integrated in either the southeast-oriented or northeast-oriented lobes (Plinian or co-ignimbrite). By plotting the natural logarithm of the isopach thickness against the square root of the isopach area, we determine deposit-thinning trends that we use to estimate total tephra fall volumes assuming an exponential decay (Pyle approach[15]; Fig. 1f, g) or a Weibull distribution function (Bonadonna and Costa approach[16]; Fig. 1f). The latter approach was not possible for the co-ignimbrite/dispersed layer as too few data points were available. The isopach-based volume calculations resulted in bulk deposit volumes of 38.6–46.3 km³ for the southeast-oriented lobe (depending on the applied thinning law) and -9.2 km³ for the northeast-oriented lobe (Fig. 1f, g). Due to the prevalent Aegean Sea wind pattern[17,18], we attribute the northeast-oriented fan lobe to the tropospheric transport of co-ignimbrite tephra from the main ignimbrite-forming phase (mainly Phase 4) and the southeast-oriented lobe to the stratospheric transport of tephra from the Plinian phase (mainly Phase 1).

To estimate DRE volumes for both lobes, we deemed cores POS513-20 and POS513-41 to be representative for proximal and distal Minoan deposits respectively and used their CT-derived mean deposit porosity estimates of 62.9% and 54.9%. We determine DRE-volumes of 3.8 ± 0.4 km³ for the northeast-oriented lobe, 17.6 ± 3.3 km³ for the southeast-oriented lobe, and 21.4 ± 3.6 km³ for the total distal fallout tephra deposits of the Minoan eruption. This is consistent with previous estimates[19,20] that indicated a tephra fall DRE-volume of 15–19 km³. We measured the volume content of lithics $lc_{vol}$ for each layer in POS513-20, resulting in 0.7 ± 0.2% for the Plinian layer (lower subunit), <0.1% for the co-ignimbrite/dispersed layer (upper subunit) and 12.4 ± 1.1% for the turbidite layer (middle subunit). The minor lithic content of both tephra layers implies a lithics-reduced DRE volume of 17.5 ± 3.3 km³ ($V_{l-red} = V_{DRE}*(1 - lc_{vol})$) for the southeast-oriented lobe, and 21.4 ± 3.6 km³ for the total distal fallout tephra deposits of the Minoan eruption (Table 1).

### Minoan ignimbrites

While Minoan tephra fall deposits (Plinian and co-ignimbrite) have been deposited hundreds of kilometres from Santorini, pyroclastic flows extended only several kilometres from the shoreline and were deposited as thick ignimbrites onshore and in the proximal marine area[21]. Marine seismic reflection profiles reveal the Thera Pyroclastic Formation (TPF[22]), which was formed by explosive volcanism on Santorini during the last 360 kyrs and has been studied in great detail onshore[5,8]. The Minoan deposits form the shallowest stratigraphic unit on Santorini and cover the Cape Riva eruption (22 ka) deposits in many areas on Santorini[5,8], including Thera's northwestern cliff (Fig. 2a). A seismic profile crossing the northern breach of the caldera wall, in direct proximity to northwestern Thera, shows two chaotic seismic units (Fig. 2b). The upper unit (yellow in Fig. 2b) is slightly thicker than the underlying unit (brown), and the boundary between them is uneven, resembling the boundary between the Minoan and Cape Riva ignimbrites on Thera (Fig. 2a). In addition, the thicknesses of both chaotic units in the seismic profiles agree with the thinning trend of the Minoan and Cape Riva ignimbrites on northern Thera (Fig. 2b). Taken together, these observations enable us to correlate the shallowest chaotic seismic units with the Minoan ignimbrites north and east of Santorini.

Defining the base of submarine Minoan ignimbrite west of Santorini is more challenging, as the TPF deposits are much thinner there (<100 ms TWT; Fig. 2d) than in the north (250 ms TWT) and it is not possible to correlate the onshore and offshore sequences. However, the Christiana Basin southeast of Santorini is affected by rifting, offsetting the internal stratigraphy of the TPF by up to 60 ms TWT (-50 m in Fig. 2d, *enlargement*). The offset increases with depth due to repeated fault activation over the last 360 kyrs, while the shallowest unit

**Table 1 | Volume estimates for the various components of the Minoan eruption deposits**

| | Bulk volume | | Porosity | | Total DRE volume | | | | Lithics content | | Lithics volume | | | | Lithics-reduced DRE volume | | | |
|---|---|---|---|---|---|---|---|---|---|---|---|---|---|---|---|---|---|---|
| | Min | Max | Min | Max | Min | Max | Mean | +− | Min | Max | Min | Max | Mean | +− | Min | Max | Mean | +− |
| Plinian tephra fall | 38.6 | 46.3 | 55% | 63% | 14.4 | 20.9 | 17.6 | 3.3 | 0.50% | 0.90% | 0.07 | 0.19 | 0.1 | 0.1 | 14.2 | 20.8 | 17.5 | 3.3 |
| Co-ignimbrite tephra fall | 9.2 | 9.2 | 55% | 63% | 3.4 | 4.1 | 3.8 | 0.4 | 0.10% | 0.10% | 0.003 | 0.004 | 0.0 | 0.0 | 3.4 | 4.1 | 3.8 | 0.4 |
| Total tephra fall | | | | | 17.8 | 25.0 | 21.4 | 3.6 | | | 0.1 | 0.2 | 0.1 | 0.1 | 17.6 | 24.9 | 21.3 | 3.6 |
| Onshore deposits | 1.4 | 1.4 | 57% | 73% | 0.4 | 0.6 | 0.5 | 0.1 | 6% | 11.7% | 0.02 | 0.07 | 0.0 | 0.0 | 0.3 | 0.6 | 0.4 | 0.1 |
| Offshore ignimbrite | 14.0 | 15.7 | 48% | 68% | 4.5 | 8.2 | 6.3 | 1.8 | 13% | 26% | 0.6 | 2.12 | 1.4 | 0.8 | 3.3 | 7.1 | 5.2 | 1.9 |
| Turbidite | 0.3 | 0.3 | 53% | 66% | 0.1 | 0.1 | 0.1 | 0.0 | 11.3% | 13.5% | 0.01 | 0.02 | 0.0 | 0.0 | 0.1 | 0.1 | 0.1 | 0.0 |
| Total flow deposits | | | | | 5.0 | 8.9 | 6.9 | 2.0 | | | 0.6 | 2.2 | 1.4 | 0.8 | 3.7 | 7.8 | 5.8 | 2.0 |
| Caldera infill | 9.0 | 10.9 | 33% | 45% | 5.0 | 7.3 | 6.1 | 1.2 | 13 % | 26% | 0.64 | 1.9 | 1.3 | 0.6 | 3.7 | 6.4 | 5.0 | 1.3 |
| Total deposit volume | 72.5 | 83.8 | | | 27.7 | 41.2 | 34.5 | 6.8 | | | 1.3 | 4.3 | 2.8 | 1.5 | 25.1 | 39.1 | 32.1 | 7.0 |
| Total collapse volume | | | | | 32.0 | 34.3 | 33.1 | 1.2 | | | | | | | | | | |

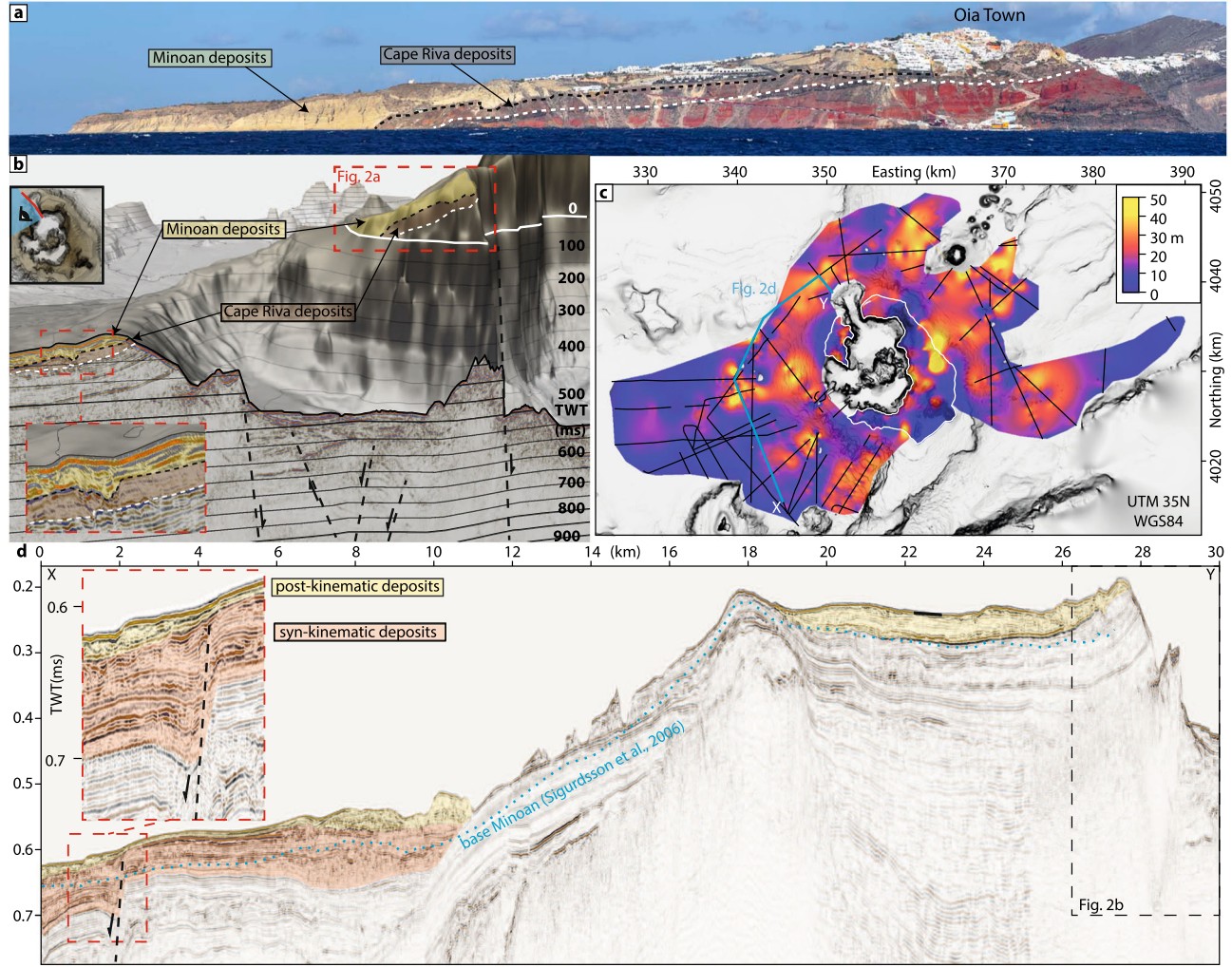

**Fig. 2 | Marine ignimbrites of the Minoan eruption. a** Photograph of the eastern cliff of Thera from a viewpoint on Therasia with an interpretation of the cliff stratigraphy[5,8] showing Minoan deposits atop Cape Riva eruption deposits. **b** Seismic profile combined with a topographic grid of Santorini, including enlargement illustrating the stratigraphic relationship between Minoan and Cape Riva deposits. The viewpoint is also from Therasia. Interpreted seismic units and corresponding subaerial cliff stratigraphy are coloured. **c** Map showing the ignimbrite thickness from seismic mapping combined with onshore deposit mapping[3]. The white line indicates the Santorini coastline. **d** Seismic profile showing the seismic stratigraphy northwest of Santorini. Enlargement shows the Christiana Fault with undeformed, post-kinematic (yellow) deposits interpreted as originating from the Minoan eruption, and offset, syn-kinematic sediments (orange) interpreted as pre-Minoan.

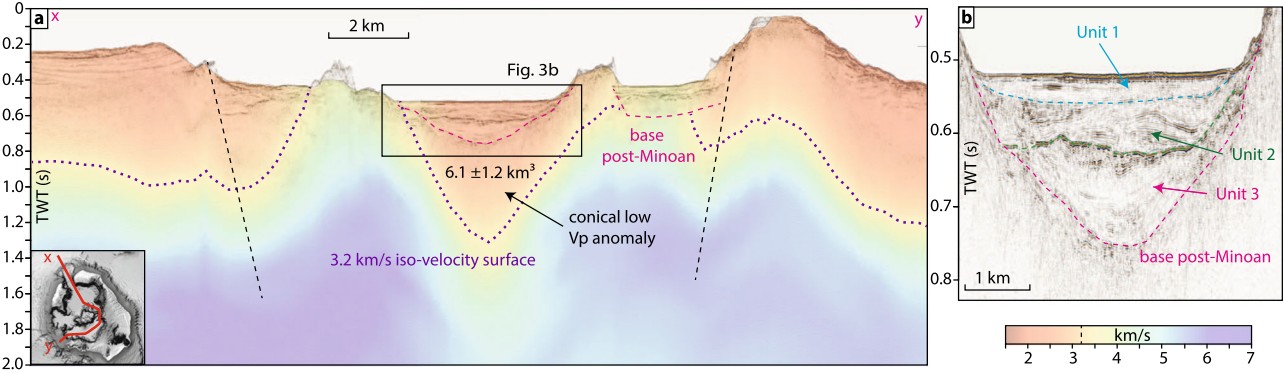

**Fig. 3 | P-wave tomography and reflection seismics. a** Seismic profile crossing the Santorini caldera through the northern breach into the northern basin, the western basin, and the south-western breach. The semi-transparent colours overlaid are from a 3D tomographic $V_p$ model[27]. Dashed pink line marks the base of Unit 3, interpreted as the boundary between Minoan and post-Minoan deposits. Dotted purple line is the 3.2 km/s iso-velocity surface, interpreted as the boundary between brecciated and volcanoclastic caldera infill and denser basement. The seismic profile does not cross the centre of the low-Vp anomaly, which is located close to Nea Kameni, and thus does not show its full extent. **b** Enlargement showing the three post-Minoan units in the northern basin.

(yellow in Fig. 2d) is unaffected by seismically resolvable displacement. This observation enables us to define the youngest (post-kinematic) stratigraphic unit west of Santorini, which includes the Minoan ignimbrite. It is not possible to constrain whether the post-kinematic deposits also include those of the Cape Riva eruption. Mapping the shallowest stratigraphic unit around Santorini and depth converting with a seismic velocity of $1.78 \pm 0.1$ km/s (ref. 22) yields the thickness map in Fig. 2c, indicating a total bulk deposit volume of 14–15.7 km³ for the offshore domain. Given the uncertainty in identification of the units, this volume is a maximum estimate.

Onshore deposits of the four phases of the Minoan eruption have been mapped on the Santorini archipelago, resulting in deposit thickness maps for all four phases[4]. Digitising the thickness contours results in bulk volumes of 0.28 km³ for the first phase, 0.34 km³ for the second phase, 0.45 km³ for the third phase, 0.36 km³ for the fourth phase, and a total bulk volume of 1.43 km³. Minoan ignimbrites deposited offshore are considered to consist primarily of material produced during Phase 4, as deposits from the earlier phases thin-out towards Santorini's coastline. Onshore ignimbrites from Phase 4 have a lithics content $lc_{wt}$ of 40–50 wt%, which is equivalent to a lithics volume content $lc_{vol}$ of 13–26 vol%, assuming that the lithics-reduced ignimbrite matrix of Phase 4 material consists of pumice (>2 mm) with an average vesicularity of ~78% (refs. 7,23). This defines the maximum lithics-reduced ignimbrite matrix porosity $\Phi_{matrix}$ for the marine ignimbrites. Our marine sediment core analysis of the co-ignimbrite tephra fall indicates an ignimbrite matrix porosity $\Phi_{matrix}$ of ~65% showing no lithics (Fig. 1a), which defines the minimum lithics-reduced ignimbrite matrix porosity for the marine ignimbrites. Assuming that the marine deposited ignimbrites were deposited by Phase 4 pyroclastic flows, and assuming a lithics volume content $lc_{vol}$ of 13–26% (previous estimates[24] suggested 8–20%), reduces the bulk porosity ($\Phi_{bulk} = \Phi_{matrix}*(1 - lc_{vol})$) for the marine ignimbrites to 48–68%. This results in a total DRE-volume of $6.3 \pm 1.8$ km³ and a lithics-reduced DRE-volume of $5.2 \pm 1.9$ km³ for the marine emplaced ignimbrites, as well as a total DRE-volume of $0.5 \pm 0.1$ km³ and a lithics-reduced DRE-volume of $0.4 \pm 0.1$ km³ for the combined onshore deposits (Table 1).

**Minoan caldera infill**
Previous studies[25–27] identified three seismo-stratigraphic units within the caldera (Fig. 3b). While the upper two units have been consistently interpreted as post-Minoan deposits, the lowermost Unit 3 has been interpreted as either down-faulted Minoan deposits[26] or as post-Minoan deposits from the northern breach collapse[27]. Our new high-resolution data show that the base of Unit 3 defines the boundary between coherent stratified sediments and an incoherent, acoustic basement, and has a funnel-shaped morphology in the northern basin of the caldera (Fig. 3). Unit 3 shows no indications for internal faulting but reveals on-lapping layers that fill up the funnel-shaped depression beneath the northern basin. This depression correlates spatially with a low-velocity anomaly in $V_p$-tomography data that formed as a result of the Minoan eruption[28], indicating that Unit 3 was deposited after the collapse and that its base represents the boundary between Minoan and post-Minoan caldera infill.

The seismic reflection data cannot properly image beneath the base of Unit 3. However, $V_p$-tomography data enable us to reconstruct the seismic velocity distribution of the upper crust beneath the caldera[28]. The $V_p$-tomography reveals a conical, low-$V_p$ anomaly with a diameter of ~3.0 km extending to ~2 km depth (Fig. 3a). Inherent to the method, the boundary of this low $V_p$ anomaly is not sharp, but 3.2 km/s has been shown to be a well-suited threshold value to distinguish volcanoclastic caldera infill from the metamorphic basement[28]. The material underneath the velocity anomaly likely consists of dense volcanic rocks, while the collapse structure itself is bound laterally by metamorphic basement[28]. Minoan ignimbrites on Santorini's flank have a $V_p$ of 1.78 km/s, so choosing 3.2 km/s as the threshold value likely overestimates the base of Minoan intra-caldera deposits. With this approach, we estimate a total caldera infill volume of ~20.6 km³ (including all Minoan and post-Minoan material between the 3.2 km/s iso-velocity surface and the seafloor). To calculate the Minoan caldera infill volume, we subtracted a depth-converted grid of the base of the post-Minoan deposits (base of Unit 3) from the 3.2 km/s iso-velocity surface, which results in a bulk volume of 9.0–10.9 km³. The intra-caldera deposits have likely been compacted, welded, or hydro-thermally altered, and thus applying porosities defined for surface deposits is not feasible. Previous analyses suggested porosity values between 33 and 45% for these deposits[28], which yields a DRE-volume of $6.1 \pm 1.2$ km³. There is no direct information about the lithics content of the caldera infill, but assuming 13–26% as for Phase 4 ignimbrites would result in a lithics-reduced DRE-volume of $5.0 \pm 1.3$ km³ (Table 1).

**Caldera collapse reconstruction**
The Santorini caldera has been shaped by at least four caldera-forming eruptions over the last 200,000 years[5]. Some cliffs surrounding the northern caldera basin predate the ~1600 BCE eruption, as indicated by a sparse, local cover of Minoan deposits, while the southern and south-eastern cliffs are morphologically fresh and likely formed during the Minoan eruption[8] (Fig. 4a). This is in agreement with cosmic-ray exposure dating, which indicates that the northern caldera wall existed before the Minoan eruption[29]. The same technique suggests that the northern caldera breach is a long-lived feature, and that the pre-

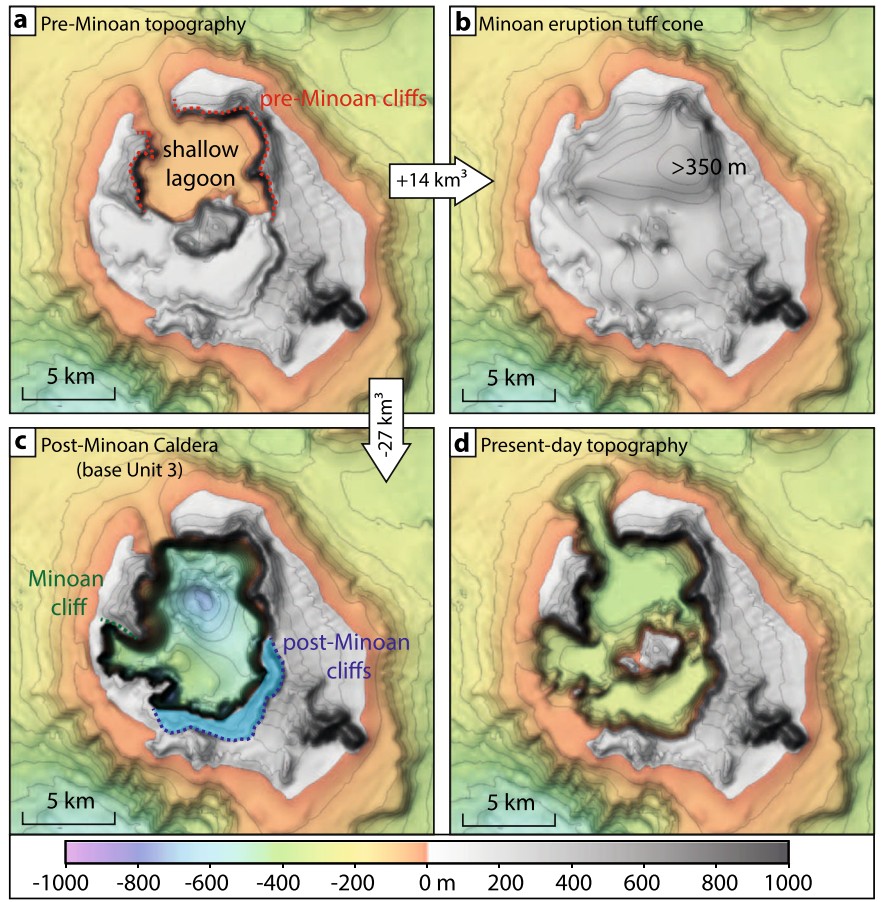

**Fig. 4 | Interpreted topographic evolution of the Santorini caldera during the Minoan eruption. a** Pre-Minoan topography with shallow marine northern caldera basin ("shallow lagoon") and pre-Minoan cliffs (dotted red lines)[8,28]. **b** Tuff cone formed during the third phase of the Minoan eruption[4,7]. **c** Post-Minoan caldera based on mapping of the base of Unit 3 (Fig. 3) with caldera cliffs ("post-Minoan cliffs") that collapsed after the caldera collapse (marked blue)[8]. **d** Present-day topography.

Minoan caldera may have been connected to the sea, while the southwestern breach and the western basin were only formed during the Minoan eruption[29]. The Minoan deposits contain lithic fragments of various origins, including geochemically distinctive andesite fragments interpreted as pieces of a pre-Minoan volcanic edifice present inside the ancient caldera[24,30]. The additional presence of stromatolite fragments in the Minoan tuffs may indicate the presence of a shallow marine lagoon in the pre-Minoan caldera[31]. Jointly, these observations enable us to define the approximate shape and size of the pre-Minoan caldera[6] (Fig. 4a). It has been suggested that, during the 3rd phase of the Minoan eruption, this caldera was filled in by a 32–42 km³ tuff cone before the caldera collapsed anew[3] (Fig. 4b). However, based on our paleo-topographic reconstruction, as little as 14 km³ of tuff would have been sufficient to fill up the northern basin, corresponding to a DRE-volume of 4.9 km³ assuming a deposit porosity of 65%. This DRE-volume agrees with our tomography-based Minoan eruption caldera infill reconstruction (6.1 ± 1.2 km³), suggesting that the entire tuff cone could have collapsed, forming the caldera infill. Using the base of Unit 3 as the post-Minoan caldera topography (Fig. 4c), we constrain the increase in caldera volume during the Minoan eruption to be ~27 km³, which, when combined with the caldera infill, yields a total caldera collapse volume of 33.1 ± 1.2 km³ (Table 1).

## Discussion
Combining our DRE-volume estimates for the turbidite layer in the marine sediment cores (-0.1 km³), the onshore deposits (0.5 ± 0.1 km³), and the marine ignimbrites (6.3 ± 1.8 km³), yields a total pyroclastic flow deposit volume of 6.9 ± 2 km³ for the Minoan eruption (Fig. 5 and Table 1). Our estimate is much smaller than the 41 km³ DRE estimated from previous seismic mapping[21]. We attribute this large discrepancy to (i) limited resolution of previous seismic data, which led to the erroneous interpretation of large parts of the TPF in the Christiana Basin as Minoan deposits (see dotted line in Fig. 2d), and (ii) applying a deposit porosity of only 25% compared to the 48–68% we used. Our estimate is also significantly smaller than ignimbrite quantifications based on mass-balance calculations[19,20] using co-ignimbrite tephra volumes in sediment cores, which implied DRE-volumes of 10–20 km³. These estimates were based on attributing 85–98% of the total mapped tephra volume to the ignimbrite phase, and quantifying the fine-depleted ignimbrite volume based on mass-balance calculations, which assumed mass ratios between ignimbrite and co-ignimbrite tephra of 41:59–54:46 and co-ignimbrite tephra volumes of 15–20 km³ DRE[19,20]. Our sediment core analysis shows a co-ignimbrite tephra volume of just 3.8 ± 0.4 km³ DRE, resulting in an ignimbrite to co-ignimbrite tephra volume ratio between 72:28 and 54:46, which is slightly elevated towards the fine-depleted ignimbrite. While our total tephra fall estimate of 21.4 ± 3.6 km³ DRE (Fig. 5) is in line with previous Minoan fall volume estimates of 15–19 km³ DRE[19,20] (Fig. 5), it indicates that the volume of Plinian and Phreatoplinian products (Phases 1–3) has been previously underestimated. An increased volume for these earlier phases of the eruption fits well with the formation of the voluminous Phase 3 tuff cone, which was likely accompanied by significant ash production.

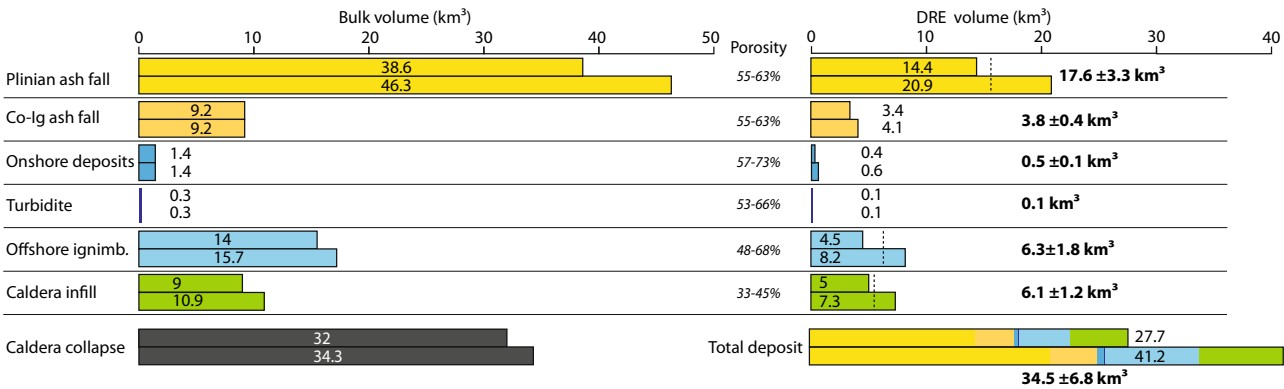

**Fig. 5 | Volumetric analyses results for the Minoan eruption.** Minimum and maximum estimates for the bulk and DRE-volumes for the various components of the Minoan eruption deposits. Conversion between bulk and DRE-volumes is based on porosity estimates for each component. Similar caldera collapse and total DRE-volumes indicate an internal consistency.

Our total eruption volume estimate of $34.5 \pm 6.8$ km³ DRE (or 27.7–41.2 km³ DRE) is in line with Pyle's (1990) estimate of 28–29 km³, which, compared to our calculation, overestimated the ignimbrite phase while not specifically considering the caldera infill (Fig. 5). Our caldera infill calculation of $6.1 \pm 1.2$ km³ DRE is the least well constrained of our estimates due to uncertainties in defining the base of the Minoan caldera infill and not being able to distinguish between Minoan material and material from earlier collapse events. Nonetheless, it is more robust than the previous estimate of 18–26 km³ DRE[3], which was based on a conceptual reconstruction of the Minoan tuff cone that could not be verified in a seismic-based follow-up study[26]. We measured or estimated the lithics content for each subunit resulting in a total lithics volume of $2.8 \pm 1.5$ km³ DRE and a lithics-reduced DRE-volume of $32.1 \pm 7$ km³ DRE (Table 1). Additional uncertainties in our total volume estimate are (i) the possibility that the mapped ignimbrite in Christina Basin may include earlier ignimbrites of the TPF and failed seafloor material, (ii) an underestimate of ultra-distal ash, (iii) distally deposited pumice rafts, and (iv) post-eruptive erosion. A ~25.7 km³ turbiditic volcanoclastic megabed in the Cretan Basin has been attributed to the Minoan eruption[32]. However, neither seismic data nor sediment cores show indications for such a large mass-transport, making this interpretation highly unlikely. In contrast, our reconstruction of the caldera collapse volume of $33.1 \pm 1.2$ km³ (Fig. 5) demonstrates an internal consistency between two independent approaches, implying high confidence in our estimates. A similar volume consistency has been noted for the ~8.9 km³ of submarine-emplaced deposits of the 1883 Krakatau eruption and the associated ~9 km³ caldera volume[33]. However, our analysis represents a significantly more detailed and error-constrained approach, by combining a unique dataset of measurements over various scales, from microscopic (sub-millimetre resolution of CT-scans) through (multi)-meter (core stratigraphy and multi-channel seismic data) to shallow crustal scales (P-wave tomography).

Our study yields a marine ignimbrite volume of just 14–15.7 km³ ($6.3 \pm 1.8$ km³ DRE), significantly smaller than previous estimates of 10–41 km³ DRE[19–21], which has broad implications for tsunami hazard assessment. The Minoan marine ignimbrite volume is comparable to the 6.5 km³ DRE emplaced by the 1883 Krakatau eruption over the course of several hours, which triggered destructive tsunami waves affecting the surrounding coasts[33,34]. Tsunami genesis during both eruptions, and by the marine emplacement of pyroclastic flows in general, are not well understood. Previous numerical simulations of the Minoan tsunami could reproduce reconstructed tsunami run-up heights around the Southern Aegean Sea[10,35], but these used tsunami source parameters that are not in agreement with our mapping. Regardless, this reduced volume estimate is still large compared to other volcanic flows, like those of historic pyroclastic flows and volcanic debris avalanches. For example, a dome collapse event on Montserrat in 2003 produced pyroclastic flows over several hours[36], with the largest having a bulk volume of 0.016 km³. The flow entered the sea, triggering a tsunami that was 4 m high on Montserrat's coast and up to 1 m at Guadeloupe[36]. In addition to the explosive blast, the emplacement of pyroclastic flows may have been a major contributor to tsunami genesis during the 2022 eruption of Hunga Tonga-Hunga Ha'apai[37]. During the 2018 Anak Krakatau sector collapse, a debris avalanche of only 0.3 km³ triggered a tsunami up to 13 m high at the coasts of Sumatra and Java, causing 437 fatalities[38]. These two events highlight the tsunamigenic potential of shoreline crossing volcanic mass-transport events. Numerical simulations of the 1888 Ritter Island sector collapse revealed that water displacement at the sea surface dominates the tsunami potential of volcanic debris avalanches[39]. It appears likely that this is also true for ground-bound pyroclastic currents, meaning that marine emplaced ignimbrites may have a greater tsunamigenic potential than previously assumed.

To date, Santorini is one of the few large-scale Holocene eruptions with a complete coverage of all eruption products. Comparatively complete records are only available for the (mainly) onshore eruption products of the (pre-Holocene) Campanian Ignimbrite eruption[40] and the 7000 BP eruption of Mt Mazama[41], for which no intra-caldera deposit measurements are available. For most eruptions of coastal or island volcanoes, including Tambora (1815), Kuwae (1425) and Samalas (1253), submarine deposit records are incomplete or nonexistent[42–45]. While tephra volumes can often be approximated based on onshore tephra thickness measurements[44,45], reliable ignimbrite volume estimates depend upon the availability of high-resolution seismic reflection imaging. In the absence of such data, total volume estimates have large uncertainties. This limitation is also exemplified by the 1815 eruption of Tambora, where offshore ignimbrite volume estimates vary greatly between 2.8 km³ DRE[43] and 21 km³ DRE[45]. To date, most of the largest and most prominent Holocene eruptions, including Samalas (1257), Kuwae (1425), Tambora (1815) and Krakatau (1883), lack geophysical (marine) ignimbrite measurements. In the case of Santorini, our results reduce the ignimbrite, caldera infill and total volume estimates significantly, highlighting the large uncertainties introduced by not constraining submarine deposit volumes.

Empirical analyses suggest a correlation between caldera area and the associated eruption volume[46]. The Santorini caldera (~83 km²) is significantly larger than the calderas of Kuwae (~56 km²), Tambora (~37 km²), and Samalas (~29 km²), while volume estimates for their associated caldera-forming eruptions[42–45] are all larger than our revised Minoan eruption estimate. In light of the large uncertainties of volume estimates without seismic datasets and the volume revision for the

Minoan eruption, our analysis indicates that eruption volume estimates mainly based on the caldera area have large uncertainties. The multicyclic nature of many calderas, including Santorini, may have an impact on the reliability of caldera area versus eruption volume correlations. This highlights the necessity of integrated seismic ignimbrite and core-based tephra analyses for other large-scale eruptions. For future analyses of major eruptions, our revised reconstruction of the Minoan eruption provides a framework for estimating eruption volumes. Only improved eruption volume reconstructions can allow reliable hazard and risk assessments for major volcanic eruptions[2].

## Methods

### Marine sediment core sampling and logging

In 2017, 47 marine gravity cores were recovered during research cruise POS513 on board RV POSEIDON from the Aegean Sea at water depth between 100 and 1200 m[12,13]. Sediment recovery in gravity cores was 100% and thickness measurements have an uncertainty of <1 mm. The sediment cores were visually described for lithological and sedimentary parameters including lithological components, colour, sedimentary structures, and occasionally drilling disturbances. Lithological components comprise tephra particles (glass, minerals), rock fragments, and microfossils. The tephra was wet-sieved into 63–125 μm and >125 μm fractions, which were used for geochemical microanalyses[12,13]. Based on the CT-scans, it was possible to define three facies within the Minoan tephra layer (tephra layer 1, flow-derived layer, and tephra layer 2). All cores were analysed for the occurrence and thickness of these layers (Supplementary Note 1 and Supplementary Dataset 2). The tephra layer distribution was used to define isopach maps, which have been used to calculate bulk deposit volumes[15,16].

### Computed tomography sediment core analysis

To better constrain the thickness, density, and porosity estimates of marine Minoan tephra, we CT-scanned two 1 m long core sections that were collected during the 2017 cruise of RV Poseidon. POS 513-20, which contains a proximal tephra deposit, and more distal POS 513-41 (refs. 12,13). To do so, we used the ProCon CT-ALPHA CORE scanner at the EARTHLAB sediment facility of the University of Bergen. By capturing 65536 grayscale values at micrometer scales, this bespoke system can resolve down-core variations in the density, porosity, and composition of marine tephra deposits with high precision[47]. Core sections were scanned at 950 μA and 145 kV with an exposure time of 334 ms capturing 2400 projections per rotation, producing imagery with a voxel size of ~100 μm. To reduce the impact of beam hardening effects, we used a 1 mm copper filter. After scanning, CT projections were reconstructed for 3-D analysis with the Fraunhofer Volex 10 software. We then relied on version 9 of the Thermo Scientific Avizo suite for image to (i) assess the volume of tephra layers, (ii) resolve down-core porosity and density variations, and (iii) aid identification of eruption phases. For this purpose, we applied the protocols outlined by (ref. 48) and (ref. 49). First, we calculated down-core CT grayscale variations—which primarily capture density differences and thus capture changes in porosity and sediment composition[50]—for 0.3 cm³ volumes at 500 μm intervals. Next, the CT density range of reconstructed scans was manually segmented to highlight air-filled pores. We then assigned a single value to this thresholded density range, before calculating down-core pore volumes at 500 μm intervals. To help disentangle the imprint of changes in density, sediment composition, and porosity on CT grayscale variability, we validated our scans against physical grain size and density measurements (Supplementary Note 2). To warrant measurement intercomparability, we targeted the same core coordinates and volumes used for CT analysis. First, 0.3 cm³ of wet sediment was extracted using a 1cc syringe. We did so at regular 1–2 cm intervals, with the exception of the particularly coarse (gravel-sized) Plinian tephra deposit between 42 and 64 cm depth in core POS

513-20 (refs. 12,13). All samples (n = 83) were subsequently dried overnight at 105 °C to determine Dry Bulk Density (DBD) as outlined by Dean Jr (ref. 51). We then prepared this material for grain size analysis on a Malvern Mastersizer 3000 by removing all organic material through loss-on-ignition (LOI) at 550 °C after van der Bilt, et al. (ref. 52). Each sample was measured 5 times to monitor analytical precision, including only averages that were reproducible (n = 34) according to ISO 13320 standards. For these grain size distributions, metric Folk and Ward measures were calculated using the GRADISTAT software by Blott and Pye. Finally, we performed a few basic geostatistical approaches in version 4 of the PAST software to assess the relation between CT-derived and physically measured estimates of density or porosity. These comprise linear regression, Gaussian smoothing, and (cross)correlation analyses (Supplementary Note 2).

### Reflections seismic analysis

The seismic database contains 650 km of high-resolution multi-channel reflection seismic profiles collected during research cruise POS538 on board R/V Poseidon in October 2019 (ref. 53). As seismic source, we used a GI-Gun in harmonic mode with a primary (Generator) and a secondary (Injector) volume of 45 in³, which was operated with a shot interval of 4 s resulting in a shot-distance of ~6 m. Seismic signals were recorded by a Geometrics GeoEel digital streamer with an active streamer length between 190 m and 250 m. The processing flow included source-receiver geometry corrections, trace-editing, frequency filtering (15–1500 Hz), surface-related multiple elimination, spherical divergence correction, time-variant frequency filtering, pre-stack time migration, top-muting, and white-noise removal, bandpass frequency filtering, normal move-out correction, stacking, trace-interpolation, and 2D time migration. The channel spacing of 1.56 m and the comparably short shot interval yielded a horizontal seismic resolution of 1.56 m and the seismic source's main frequency of 125 Hz resulted in a vertical resolution of 4-8 m at the seafloor (using the λ/4- or λ/2-approximation and a seismic velocity of 1.6 km/s). These datasets were combined with previously collected multi-channel[25] and single-channel seismic data[21] and analysed using the seismic interpretation software suites Kingdom Suite by IHS Markit and Petrel by Schlumberger. We defined the base of the Minoan ignimbrite as described in the main text and used the seafloor reflection as its top. Both horizons were subtracted and the TWT-differences was converted to thickness using seismic velocities of 1.68 km/s as a minimum estimate and 1.88 km/s as a maximum. The offshore deposit thicknesses were combined with an onshore deposit thickness map[3] and gridded. These grids were used to calculate the deposit bulk volume by calculating the average grid value (thickness) and multiplying it by the grid area. The same seismic dataset was used to map the base of Unit 3 within the caldera.

### P-wave tomography

The $V_p$-tomography dataset was acquired on board R/V Marcus Langseth in 2015 using 90 ocean-bottom and 65 land seismometers and a 3600 cubic inch airgun array[28]. About 14,300 controlled-sound sources covered an area of 120 × 60 km around Santorini. First arrivals of crustal refractions (Pg) were picked on hydrophone, vertical, or a stack of the hydrophone and vertical channels and integrated into a Pg travel time inversion to calculate a $V_p$-tomography model (ref. 28). Finally, the tomography was converted from the depth into the time domain to be integrated with the reflection seismic dataset using the seismic interpretation software Petrel.

## Data availability

Tephra thickness measurements for the marine sediment cores from research cruise POS513 are available at https://doi.pangaea.de/10.1594/PANGAEA.937928. The seismic reflection profiles obtained doing POS538 are available at https://doi.pangaea.de/10.1594/PANGAEA.956579. The p-wave tomography dataset is available at https://www.

marine-geo.org/tools/files/31273. High-resolution versions of the CT scans presented in this study can be accessed at https://doi.org/10.18710/P6LWL5.

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

## Acknowledgements
We thank the masters and crews of RV Poseidon for their support during research cruises POS512 and POS538. We also thank Schlumberger and IHS Markit for granting educational licenses to their software. Jonas Preine's contribution was supported by the German Research Foundation DFG (HU698/25). Tim Druitt's contribution is part of the laboratory of Excellence ClerVolc (Laboratory of Excellence ClerVolc Contribution Number 585). Willem van der Bilt`s contribution was supported by a Starting Grant (TMS2021STG01) from the Trond Mohn Stiftelse (TMS). We would like to thank Ali Bolkar Kiliç, who carried out density and grain size measurements on the sediment cores in the EARTHLAB facility of the University of Bergen.

## Author contributions
J.K., C.B., and C.H. were responsible for organising and designing the reflection seismic experiments. J.K, J.P, G.J.C., F.S., P.N., M.K., and J.E. acquired, processed and interpreted the reflection seismic data. S.K. collected the sediment cores, W.G.M.v.d.B. and J.M.C. performed the CT scans and all three interpreted the sediment cores. E.E.E.H. provided and interpreted the P-wave tomography data. J.K, J.P., T.H.D., S.K., and S.C. integrated the geophysical results into the volcanological framework of Santorini. J.K, J.P, and G.J.C. drafted the paper, while all co-authors discussed the dataset and provided comments and corrections to the paper.

## Funding

## Competing interests
The authors declare no competing interests.
