## [Peer Review File · Nature Communications]

Revised Minoan eruption volume as benchmark for large volcanic eruptionsREVIEWER COMMENTS

Reviewer #1 (Remarks to the Author):

Dear Authors and Editors,

This study is a comprehensive analysis of the distal deposits of the Minoan eruption, and comprises core data and geophysics data. It combines this data, which crosses scales from the micro-m to km, with a reevaluation of some of the onshore deposits, to estimate the volume of the Minoan eruption. This is probably the most comprehensive approach to volume estimation I have seen for an eruption. The approach taken here could provide a framework for volume estimation at other volcanoes. Volume estimation is important as it provides the basis for understanding eruption scale and volcanic hazard assessment. Thus I expect the results to be of significance to the volcano community in general. Given the historical importance of the Minoan Eruption, it may also prove significant to other fields such as archaeology.

In general, this is a well constructed manuscript which draws the reader through their analysis. Broadly, the methods are appropriate and applied well, in a somewhat novel approach. Enough detail is provided that those wishing to replicate the study can. The analysis is presented well and interpretations follow logically from the data presented. I have some comments about the manuscript which I think will need to be addressed before publication.

For the core analysis (methods), I'd like to see a statement on the limits of this data. How many cores captured the entire sequence, to the base? And what was the % recovery? E.g. how well constrained are the thickness estimates derived from core analysis?

For the Computed-tomography sediment core analysis (methods) I feel like a reference to S2 might be helpful? I'd also like to see a clearer estimate of the error range in porosity and therefore density in this section.

I am somewhat unconvinced by the isopach maps presented in Figure 1. There are too few data points on Fig 1 for this to be convincing. How are the 1, 3 and 6 isopachs constrained? there doesn't seem to be any datapoints beyond the circle between 6 and 8? This point, which is the furthest NE datapoint, is arguably within the datacloud of points that constrain the red isopach lines. Furthermore, the datapoints do not have any values, which leaves the reader unclear on the certainty/uncertainty with which these lines are drawn. I suggest that a more detailed isopach map is presented in the supplementary materials, which includes the thicknesses, with a more nuanced discussion on the data

that supports it and the uncertainty involved. This could perhaps be added to the map presented in S2? Or a separate figure in an additional supp mat.

In lines 108-112 the authors state "The thickness distribution of the Plinian layer (lower subunit) agrees well with the thinning trend seen in more distal sediment cores from the Eastern Mediterranean¹¹ and results in isopachs with a southeast-oriented deposit lobe. The coignimbrite/dispersed layer (upper subunit) follows the trend of thickness constraints from Aegean islands, mainland Turkey and the Black Sea¹⁴, defining a northeast-oriented lobe". Could this data be presented so the reader can be convinced of the good correlation?

I may have missed something, but in lines 134-135, why are there no error estimates for the first two values?

Some minor points to follow.

Line 50 - should the DRE acronym be defined at first use given the general readership of Nat Comms?

Line 95 - I generally tend to prefer PDC terminology, so welcome its use here. In other places within this manuscript however, the authors use pyroclastic flow and pyroclastic surge. I'd favour a consistent adoption of PDC through out. At least the manuscript should be internally consistent with its use of terminology.

The figures are generally very good. I'd however like to see them passed through a colour blind checker because there is some use of green/red next to each other that some readers may find difficult to distinguish.

Line 399 - "for 0.3 cm³ volumes at" needs superscript.

Given the importance of this work, I'd like to see full publication of the datasets that underpin this work, such as the seismic lines, p-wave tomography and high resolution core scans. I would like to see these published alongside the manuscript and not 'data available on request' which normally means that the data remains inaccessible to future research.

Best wishes and good luck in progressing this manuscript.

Rebecca Williams

Reviewer #2 (Remarks to the Author):

General comments

My general comments are guided by the questions below provided in the guidelines for referees.

What are the noteworthy results?

The work by Karstens and co-workers provides a revised eruption volume for the large-magnitude Late Bronze Age (Minoan) eruption of Santorini in Greece. Previous estimates of the volume of the Minoan eruption vary widely, with the new estimate presented here (31.4 ± 5.2 km³ DRE (dense-rock equivalent) volume) being near the lower limit of these previous estimates (c. 19 km³ DRE) and significantly lower than the maximum volume estimates proposed previously (c. 86 km³ DRE). The large differences in the previous eruptive volume estimates largely stem from uncertainties in the eruption's volume of ignimbrite and the generation mechanism of the distal ash fall (Plinian vs co-ignimbrite ash), which are largely deposited offshore. In this work, the authors use seismic reflection profiles and observations from marine sediment cores to better constrain the volumes of these deposits in the marine realm and, as such, provide the so far most robust and best constrained bulk (i.e. tephra) volume estimate of the Minoan eruption.

Will the work be of significance to the field and related fields? How does it compare to the established literature? If the work is not original, please provide relevant references.

The eruptive volumes of large volcanic eruptions are notoriously difficult to determine accurately, and the problem is compounded further for eruptions, whose deposits are dispersed widely and where large volumes of the eruptive products get transported and deposited in the sea. This work provides the yet most detailed volume determination of a large volcanic eruption where most of the deposits are in the sea, exploiting data from a dense array of seismic reflection profiles and results from marine sediment cores in the area. Such data do not, or rarely exist for other large eruptions, highlighting the novelty of the work and providing a framework of what needs to be done to derive accurate eruption volumes of large-magnitude eruptions. Such information is not only important from a volcanological and hazard perspective but also for a better assessment of the environmental and climatic effects of such eruptions.

Does the work support the conclusions and claims, or is additional evidence needed?

I believe that the estimate of the bulk volume of the Minoan eruption presented here, based on the interpretation of both seismic reflection and marine sediment core data, is the most robust and accurate to date. However, there are additional complexities and uncertainties that come in when bulk tephra volumes are converted to magma volumes (i.e. DRE (dense rock equivalent) volumes). Among the main uncertainties are difficult-to-constrain “average” deposit porosities and, in particular, the proportion of non-juvenile or lithic clasts in the deposits. As it stands, and unless I missed it, the work doesn’t fully explain how lithic concentrations (i) have been estimated or considered in the DRE calculations, (ii) how they vary between the different deposits from phase 1 to phase 4 of the eruption, and (iii) how they differ as a function of (a) distance from source and (b) grain size. The work should clearly indicate how “average” lithic concentrations were derived and the uncertainties for the derived DRE volume given the expected variations in lithic concentrations both onshore and offshore. Would a more detailed incorporation of this aspect in the DRE volume determination even lead to a lower DRE volume than the $31.4 \pm 5.2 \text{ km}^3$ stated?

Are there any flaws in the data analysis, interpretation and conclusions? Do these prohibit publication or require revision?

Apart from the previous comment, I feel that the statement that the results presented for the Minoan eruption “may point to a systematic overestimation of the volume of the other large-scale Holocene eruptions” is too speculative and should be avoided, unless further support is given in favor of this statement.

Is the methodology sound? Does the work meet the expected standards in your field?

The paper provides a sound methodology in most aspects, which are provided as supplementary material. The conversion between bulk (tephra) and DRE volume should include further detail how the lithic concentrations were taken into account, and the implications for the derived DRE volume.

Is there enough detail provided in the methods for the work to be reproduced?

See above. More detail of the lithic concentrations and variations in the Minoan deposits, and how they affect the stated magma or DRE volume of the eruption should be added.

Specific comments

Linked to line numbers in submitted manuscript.

21: DRE volume = magma volume; how was the lithic clast content of the deposits handled? This should be clarified and discussed in the paper. See general comments above.

30: Replace “risk” with “hazard”?

31: “pyroclastic flows”, “ignimbrites”, “pyroclastic density currents” – use consistent terminology.

46: delete “deposits”; “ignimbrite”, per definition, is a deposit.

59: “Finally, after the connection to the sea was closed and water evaporated...”; please add some detail. How did both of these things happen?

61: What is the evidence or what are the arguments for caldera collapse both during and after the eruption? State clearly when caldera collapse is thought to have occurred.

70: Is this really the case? Perhaps “one of the first” or “the best constrained” might be better expressions here. “> 6 magnitude”; do you mean “magnitude” (value linked to mass of erupted magma) or “VEI” here? Please be clear about it in the paper.

73: “tephra” instead of “ash”.

74: Briefly specify what this unique geochemical signature is.

79/82: What do “low lithic content” and “lithic-rich” mean in terms of numbers? Values for the lithic content are important for conversion from bulk to DRE volume. Consider adding observed ranges of values here in brackets.

88: “Pronounced porosity variations”; provide range of values. See previous comment.

101: What is the significance of the stated color change? A compositional change or a change in clast types? Explain.

104: State again at the beginning of this section what units the “marine ash fall volume” refers to. The lower and upper units described before? Consider replacing “ash” with “tephra” (or, at least, use consistent terminology throughout).

106-107: How were the distinct layers identified in the marine sediment cores investigated correlated when they were “combined with more distal thickness measurements from the region”? Briefly add a comment on this issue.

116-117: How does the use of different deposit thinning laws affect the results obtained? A comment on this would be useful.

124-126: “To estimate DRE volumes for both lobes, we calculated the deposit porosity for proximal and distal marine ash layers based on cores POS513-20 and POS513-41, deemed representative for proximal

and distal Minoan deposits, respectively". Why were these deemed representative? Please add an explanation, also considering a previous statement that porosities can vary quite significantly (88).

133-137: What are the variations or standard deviations of the porosity values? How would the results be affected if the range of observed values would be considered? As a minimum, such a variation should be accounted for in any uncertainties. What is the effect of lithic concentrations on these DRE values?

140: "Minoan ash fall"; state that this is Plinian and co-ignimbrite, if correct.

144: Change "350 kyrs" to "~ 360 kyrs" (see also 159).

161-162: "It is not possible to constrain whether the post-kinematic deposits also include those of the Cape Riva Eruption". If they do, this may affect the calculated volume of the Minoan deposit here, which would therefore be a maximum.

171-176: Karatson et al. (2018) did a detailed analysis of the lithics content of the Minoan deposits on Santorini. Are these within the range given? What are the implications for the estimated DRE volume of any reduction (or variation) in lithics content with distance from source?

194: What is meant by the term "basement" here. Please clarify.

207: See Wulf et al. (2020) for latest dates, though the number given is OK in the context provided.

235-236: "ii) applying a deposit porosity of only 25% compared to the 65-78% we used". This makes a huge difference in the conversion to DRE. This means that there is an additional level of complexity and uncertainty regarding the DRE conversion and volume, despite a well-defined bulk (tephra) volume.

239: "tephra" instead of "ash"?

249: "+/- 5.2 km³"; does this mean max./min. (i.e. a range of 26.2-36.6 km³). It may be worth stating this somewhere.

280: Add "For example" at beginning of sentence "A dome collapse event...".

285-289: Could a reference to the Hunga Tonga-Hunga Ha'apai, eruption be made here?

291: I don't think the term "marine volcanoes" is adequate, as used here.

295-297: See also Self et al. (2004).

302-313: For Tambora, the volume of Self et al. (2004) is not smaller than the proposed new volume of the Minoan eruption.

335-336: It is unusual to see the different thinning behavior of the two ash layers. Is there any explanation for this and what are the implications, if any, of this different behaviour for the volume calculations?

339-341: The Cape Riva deposits could be sketched slightly more accurately in "A", as they currently also include the "Andesites of Oia" above the red deposits of the Upper Scoriae 2 eruption.

369-372: "Conversion between bulk and DRE-volumes is based on porosity estimates for each component". Add detail on how this was done in the "Methods" section, including, e.g. treatment of lithics concentrations.

Reviewer #3 (Remarks to the Author):

I have read with interest the paper Revised Minoan eruption volume as benchmark for large volcanic eruptions by Karsten et al. The paper presents a new evaluation of the volumes emplaced during the succession of phases of the Minoan eruption, based on high resolution reflection seismic survey, essential for a volcanic island where most of the deposits are offshore.

The new calculations match those already known in the literature as total volume (Pyle 1990) but show some differences for the individual phases, especially for the ignimbrite which appears slightly smaller in volume than previously assessed (however well within the order of magnitude).

The methods used are state-of-the-art and the results allow some refinement of the volume calculations for the Minoan eruption, however well within errors usually associated with these type of calculations for other ignimbrites (see discussions in Mason et al 2004 BV, Folkes et al 2011 BV, S Brown et al 2014 JAV, Silleni et al 2020 FREAR).

The results do not provide any new information about the succession of volcanic events associated with the Minoan, nor they are used to improve the understanding of processes such as the runout of ignimbrites in general (e.g. Roche et al 2021 JGR, Giordano and Cas, 2021 ESR), or on water (e.g. Dufek and Berganz 2007 GGG) or the style and timing of caldera collapse (e.g. Roche and Druitt 2001 EPSL). The claimed relevance for tsunamigenic processes is only qualitatively proposed.

In spite of the title, the manuscript as it is does not stand out as a benchmark for large volcanic eruptions.

It is rather narrow in scope and implications, although obviously of interest for those who study the Minoan eruption. The fact that this is a well known case study does not make any research on it of immediate broad relevance and interest. The need of accurate off-shore studies in partly submerged calderas is certainly not a new concept, nor this one is the first paper on such topic. For example similar studies have been done in the Campi Flegrei area (e.g. Natale et al 2022a Bas Res; 2022b J Struct Geol, and references therein).

In essence while the overall quality of data and results is good, there is an (unsubstantiated) overstatement of their relevance in the title, in the introduction and in the discussion.

The lack of any reference to the papers quoted above (or others similar), either to provide a broader view in the introduction, or to find an outcome for the new volume calculations, or else to compare the results with other case studies, further indicates that the authors have focussed on their results and have not tried to address broader issues.

This manuscript is well tailored for a specialized journal either in geophysics or in volcanology but certainly not for Nature Communications, where the reader expects to find broad-ranging studies.

Rebuttal Letter

Once again, we would like to thank Rebecca Williams and two anonymous reviewers for their helpful and constructive comments. We agree with all suggestions and we changed the manuscript accordingly. The main changes are (1) providing links to access the various datasets in the “Materials & Correspondence” and (2) shortening the abstract from 200 to 150 words to fulfil the journal requirements:

Despite their global societal importance, the volumes of large-scale volcanic eruptions remain poorly constrained. Here, for the first time, we integrate seismic reflection and P-wave tomography datasets with computed tomography-derived sedimentological analyses to estimate the volume of the iconic Minoan eruption. Our results reveal a total dense-rock equivalent eruption volume of $33.9 \pm 6.6 \text{ km}^3$, which encompasses $21.4 \pm 3.6 \text{ km}^3$ of tephra fall deposits, $6.9 \pm 2 \text{ km}^3$ of ignimbrites, and $5.5 \pm 1 \text{ km}^3$ of intra-caldera deposits. Approximately 2.7 km^3 of the total material consists of lithics. These volume estimates are in agreement with an independent caldera collapse reconstruction ($\sim 32.5 \text{ km}^3$). The results show that the Plinian phase contributed most to the distal tephra fall, and that the pyroclastic flow volume is significantly smaller than previously assumed. Our benchmark reconstruction demonstrates that complementary geophysical and sedimentological datasets are required for reliable eruption volume estimates, necessary for regional and global volcanic hazard assessments.

In the following, you can find a detailed point-to-point response to the reviewers’ comments:

Reviewer #1 (Rebecca Williams):

Thank you for the clear response to reviews and track changes document. I am largely satisfied that my comments have been considered by the authors, adequately responded to and sufficient changes to the manuscript and supplementary materials made (if light on occasion). The addition of a table of data underpinning the isopach maps is particularly welcomed.

Comment 1: One outstanding comment is regarding data availability. In the response the authors state the locations that data will be available or already is available. But this statement is not made in the manuscript itself. As it stands, the current data availability statement is not inline with Nature's policy on data availability. If accepted, the authors should deposit the seismic and core scans on PANGAEA as they state in the rebuttal and this should be detailed in the data availability section prior to the publication of this manuscript. As the P-wave tomography data is available, this section should be updated accordingly before the paper is accepted.

We fully agree that datasets should be made available and are happy to share all the presented datasets with the scientific community. We have uploaded the datasets and included following statement:

“Tephra thickness measurements for the marine sediment cores from research cruise POS513 are available at <https://doi.pangaea.de/10.1594/PANGAEA.937928>. The seismic reflection profiles obtained doing POS538 are available at <https://doi.pangaea.de/10.1594/PANGAEA.956579>. The p-wave tomography dataset is available at <https://www.marine-geo.org/tools/files/31273>. High-resolution versions of the CT scans presented in this study can be accessed at <https://doi.org/10.18710/P6LWL5>.”

Comment 2: Additionally, in response to another reviewer, the authors have inserted the sentence "In addition to the explosive blast, the emplacement of pyroclastic flows may have been a major contributor to tsunami genesis during the 2022 eruption of Hunga Tonga-Hunga Ha'apai". Whilst the reference to this eruption is useful here, the exact location of this sentence seems out of place? Perhaps it sits better following the sentence on Montserrat?

We agree and follow this suggestion.

Comment 3: Finally, some minor typos/errors may have crept in during the latest revisions. For example line 148 - change to 'a tephra'.

We have carefully reviewed our manuscript for additional typos and errors.

Reviewer #2 (Remarks to the Author):

I appreciate the constructive response of the authors to my comments and their attempt to address these. I am satisfied with the responses and the changes implemented in the revised manuscript, which I believe improved the paper overall.

Comment 4: There is one suggestion though, which I would like to see implemented in the manuscript: In the abstract, line 21, the authors state the "total eruption DRE-volume". This is fine, but it would be useful to mention also the bulk tephra volume here from which the DRE volume is derived. In addition, the "lithics-reduced DRE volume" should be mentioned in the abstract, as this is a more realistic DRE (magma) volume (or at least it should be mentioned by what percentage the lithics content reduces the "total eruption DRE-volume". The former ("total eruption DRE-volume") is a maximum DRE volume, and this should be made clear to the reader to avoid confusion and use of the values by others in the future.

We generally agree with this suggestion and we would have included the bulk volume estimates in the abstract, if the abstract length requirements would be different. However, the required substantial shortening of the abstract to 150 words makes it impossible to state all bulk volume estimates already in the abstract. However, we followed the suggestion to include the lithics-reduced volume calculation.